# Influence of the Bile Acid Transporter Genes *ABCB4*, *ABCB8*, and *ABCB11* and the Farnesoid X Receptor on the Response to Ursodeoxycholic Acid in Patients with Nonalcoholic Steatohepatitis

**DOI:** 10.3390/jpm13071180

**Published:** 2023-07-24

**Authors:** Henriette Kreimeyer, Katharina Vogt, Tobias Götze, Jan Best, Oliver Götze, Jochen Weigt, Alisan Kahraman, Mustafa Özçürümez, Julia Kälsch, Wing-Kin Syn, Svenja Sydor, Ali Canbay, Paul Manka

**Affiliations:** 1Department of Internal Medicine, University Hospital Knappschaftskrankenhaus, Ruhr-University Bochum, In der Schornau 23-25, 44892 Bochum, Germanyjan.best@kk-bochum.de (J.B.); oliver.goetze@kk-bochum.de (O.G.);; 2Department of Gastroenterology and Hepatology, University Hospital Essen, Hufelandstraße 55, 45147 Essen, Germanyjochen.weigt@med.ovgu.de (J.W.);; 3Department of Gastroenterology, Hepatology, and Infectious Diseases, Otto-von-Guericke-University Hospital Magdeburg, Leipziger Strasse 44, 39120 Magdeburg, Germany; 4Division of Gastroenterology and Hepatology, Department of Internal Medicine, Saint Louis University School of Medicine, St. Louis, MO 63104, USA; 5Department of Physiology, Faculty of Medicine and Nursing, University of Basque Country UPV/EHU, 48940 Leioa, Spain

**Keywords:** NASH, UDCA, bile acid transporter, ALT, GGT, TE

## Abstract

The prevalence of NAFLD and NASH is increasing worldwide, and there is no approved medical treatment until now. Evidence has emerged that interfering with bile acid metabolism may lead to improvement in NASH. In this study, 28 patients with elevated cholestatic liver function tests (especially GGT) were screened for bile acid gene polymorphisms and treated with UDCA. All patients had a bile acid gene polymorphism in *ABCB4* or *ABCB11*. Treatment with UDCA for 12 months significantly reduced GGT in all patients and ALT in homozygous patients. No difference in fibrosis was observed using FIb-4, NFS, and transient elastography (TE). *PNPLA3* and *TM6SF2* were the most common NASH-associated polymorphisms, and patients with *TM6SF2* showed a significant reduction in GGT and ALT with the administration of UDCA. In conclusion, NASH patients with elevated GGT should be screened for bile acid gene polymorphisms, as UDCA therapy may improve liver function tests. However, no difference in clinical outcomes, such as progression to cirrhosis, has been observed using non-invasive tests (NITs).

## 1. Introduction

Non-alcoholic fatty liver disease (NAFLD) has a prevalence of nearly 25% worldwide [1,2,3] and eventually progresses to non-alcoholic steatohepatitis, fibrosis, cirrhosis, and finally hepatocellular carcinoma (HCC). Treatment focuses on weight loss, dietary modification, and the management of comorbidities [4]. Independent risk factors for non-alcoholic steatohepatitis (NASH) progression include type 2 diabetes (T2DM), insulin resistance, and fibrosis [5,6]. However, NAFLD and NASH can also occur in lean individuals, referred to as lean NASH. In patients with NAFLD, gamma-glutamyl transpeptidase (GGT) dynamics is a biomarker of advanced fibrosis [7].

Primary bile acids (BAs), such as cholic acid and chenodeoxycholic acid, are produced in hepatocytes and secreted against a gradient into the canaliculi by the bile salt export pump (BSEP; coding gene: ATP binding cassette subfamily B member 11; *ABCB11*) in the apical membrane of hepatocytes. Secretion depends on the amount of cholesterol and lipids in the membrane, which can be altered by the flippase familial intrahepatic cholestasis-1 (FIC, coding gene: ATPase phospholipid transporting 8B1; *ATP8B1*) and the flippase multidrug resistance protein (MDR3, coding gene: ATP binding cassette subfamily B member 4; *ABCB4*). The expression of BSEP is regulated by the farnesoid X receptor (FXR, coding gene: nuclear receptor subfamily 1 group H member 4; *NR1H4*) [8]. 

The synthesis and serum levels of BAs have been found to correlate with the severity of nonalcoholic NAFLD [9]. In recent years, there has been a growing emphasis on studying BA metabolism in the context of NASH [10]. One reason for this increased focus is the promising therapeutic response observed with obeticholic acid, an agonist of the bile acid receptor FXR, in NASH treatment. Notably, interim analysis of a phase 3 study has demonstrated histological improvement in fibrosis among patients, irrespective of polymorphisms in bile acid transporter genes. FXR plays a crucial role as a transcription factor in regulating the expression of various enzymes and transport proteins involved in bile acid synthesis and transport within the liver and intestine.

Mutations and polymorphisms in genes contributing to bile acid transportation can result in progressive familial intrahepatic cholestasis (PFIC), low phospholipid-associated cholestasis (LPAC), intrahepatic cholestasis of pregnancy (ICO), or benign recurrent intrahepatic cholestasis (BRIC) [11]. PFIC is a heterogeneous entity which leads to liver cirrhosis and HCC in newborns and adults. Presently, six different types have been described with slightly different clinical phenotypes. It is differentiated between high-GGT and low-GGT entities. Roughly, PFIC 3 (polymorphism in MDR3, *ABCB4*) leads to a disturbance in phospholipid secretion and consequently to an imbalance in micelle formation and an increase in free bile acids [12,13]. These lead to hepatocyte damage and an increase in GGT. PFIC 1,2,4,5,6 result in the reduced secretion of bile acids and therefore the accumulation of hepatocytes [14]. This leads to damage of hepatocytes and increase in alanine aminotransferase (ALT), but normally low GGT levels. BRIC seems to be a less harmful phenotype of all PFIC polymorphism. LPAC and ICP are described for MDR3 polymorphism or MDR and BSEP polymorphism and can be treated with ursodeoxycholic acid (UDCA) [15]. 

Several polymorphisms have been described in MDR3-associated diseases. *ABCB4* c. 711 A>T (p.I237=, *rs2109505*) is associated with elevated ALT and GGT and a higher risk of cirrhosis and HCC [16]. *ABCB4* c. 523 T>C (p.T175A, *rs58238559*) is correlated with liver stiffness [17]. 

UDCA is mainly used in cholestatic liver diseases. It activates a typical and alternative transport mechanism for bile acids. The accumulation of bile acids in hepatocytes contributes to an up-regulation of inflammatory cytokines and hepatic cell activation, resulting in fibrosis [18].

Steatotic liver disease is the most abundant liver disease, and patients can show a cholestatic condition. As therapy has emerged, bile acid metabolism seems to be one of the key players, and empirically it has been shown that patients improve with UDCA therapy. Following the human genome project, BA transporter polymorphism has been described and linked to rare genetic diseases. Empirically, it has been observed that NASH patients with cholestatic laboratory findings benefit from UDCA therapy [19,20,21]. However, a statistical evaluation and quantification of this effect, as well as an examination of the possible influence of gene polymorphisms of the above-mentioned transporters, are not yet available. Therefore, in this retrospective analysis, we investigated the course of various liver parameters in NASH patients who underwent “off-label” treatment with UDCA because of an existing cholestatic laboratory profile and in whom gene polymorphism determination was available.

## 2. Methods

### 2.1. Patient Information, Data Collections and Ethical Considerations

This retrospective study was performed in line with the “Declaration of Helsinki” (latest revision Fortaleza, Brasil; 2013). The study was approved by the ethics committee of the University of Essen (20-9137-BO, 25 May 2020).

Twenty-eight patients (53% male, 47% female) with NASH and elevated GGT level treated at Universal Hospital Essen and Universitätsklinikum Magdeburg between 2012 and 2019 were included. Relevant clinical data were extracted retrospectively from the electronic medical record. SNP analyses (“single nucleotide polymorphism”) of the known polymorphisms of the relevant genes were performed in NASH patients with a cholestatic picture here in the context of clinical diagnostic.

NASH was diagnosed by ultrasound, elevated liver enzymes, and serologic exclusion of viral, autoimmune, and other metabolic diseases. Sequenced polymorphisms were analyzed for associations with GGT, alkaline phosphatase (ALP), transaminases, transient elastography and fibrosis scores (NFS and Fib-4). NFS and Fib-4 [22,23] were calculated as previously described. Data were compared for every polymorphism between wildtype, heterozygote, and homozygote patients.

### 2.2. Sequencing

Quantities of 5–10 mL EDTA blood was maintained from all patients and genotyping of ABCB4 (rs45575636, rs1202283, rs58238559, rs2109505), ABCB11 (rs72549402, rs497692, rs2287622), TM6SF2 (rs58542926), NR1H4 (rs56163822), ATP8B1 (rs146599962, rs34018205, rs121909100, rs765889649), and PNPLA3 (rs738409) (described in Table 1) was performed using qPCR (TaqMan fast 7500), as described previously [15,24].

### 2.3. Statistics

Analyses were performed using Microsoft Office Excel 2013, Graph Pad 9.3.0 (GraphPad Software Inc., La Jolla, CA, USA) and R (4.2.3). Differences between and in groups were analyzed using the Friedman test (paired, non-normal distribution), Wilcoxon test (paired, non-normal distribution), p-adjustment using the Bonferroni method, and a paired *t*-test. If three timepoints were compared, the Friedman test was performed, if possible, followed by pairwise comparison using the Wilcoxon test. A value of *p* < 0.05 was considered to be statistically significant and adjusted for multiple comparisons (using Bonferroni) when performing multiple tests. All comparisons are displayed as medians with a minimum and maximum range; this is stated otherwise if not. 

## 3. Results

### 3.1. Demographic Data

Due to elevated GGT levels, off-label therapy with UDCA was performed in 28 patients [42.29 yrs (17–75), 47% female]. In addition, these patients were screened for potential polymorphisms in bile acid transporter genes.

Mean BMI was 26.98 kg/m^2^ (19–39), 19 (67.8%) patients were obese or suffered from adipositas. T2DM was only present in six (21.43) patients. Eight (28.6%) patients reported moderate alcohol consumption, and one (3.6%) rare alcohol consumption. Table 2 shows detailed clinical characteristics and co-morbidities.

Noninvasive testing for fibrosis and steatosis using transient elastography was performed in 21 and 14 patients, respectively. Steatosis was measured as the controlled attenuation parameter (CAP). When applying CAP values of >260 db/m as cut-off for NAFLD and NASH patients, advanced steatosis (S3) was present in five (35.7%) patients. One (7.1%) patient demonstrated mild steatosis (S1). Five (23.8%) patients presented LSM values >9.7 kPa, indicating advanced fibrosis (F3), one (4.76) patient demonstrated moderate fibrosis (F2), and fourteen (66.7%) patients no or mild fibrosis (F0-F1).

### 3.2. High Prevalence of Polymorphism in ABCB4 or ABCB11 in NASH Patients with Elevated GGT-Levels

The genotyping of bile acid transporter gene polymorphism was performed as described above. The results are summarized in Table 3 and Figure 1. All patients were either carriers of a polymorphism in the bile acid transporter gene *ABCB4* (26 (92.9%)) or *ABCB11* (26 (92.9%)). None of the following polymorphism was detected in any patient: *ABCB4* p. R590Q (rs45575636), *ABCB4* c.523 T>C (p. T175A, rs58238559), *ABCB11* c. 1445 A>G (p. D482G, rs72549402), *ATP8B1* c. 134 A>C (p. N45T, rs146599962), *ATP8B1* c. 1286 A>C (p. E 429A, rs34018205), *ATP8B1* c. 1982 T>C (p. I661T, rs121909100).

### 3.3. Polymorphisms in PNPLA3 Were the Most Common among NASH-Associated Genes

*PNPLA3* (rs738409 C>G, p. I148M; patatin-like phospholipase domain containing 3) was present in 17 (43.5%) patients and in *TM6SF2* (rs58542926 c.449 C>T, p.Glu167Lys; transmembrane 6 superfamily member 2) in only 3 (7.7%) patients. All patients with one of these polymorphisms also carried a polymorphism in at least one other bile acid transporter gene. In patients with homozygote *PNPLA3* polymorphism, a significant decrease in ALT and GGT could be seen after 12 months. No statistical analysis was performed in *TM6SF2* polymorphism patients as only two had a heterozygous carrier status, but using descriptive analysis revealed a decrease in ALT and GGT (*ALT:* 45.5 (21–70) U/l vs. 24 (23–25) U/l; *GGT:* 299 (198–400) U/l vs. 186 (61–310) U/l).

### 3.4. UDCA Therapy Leads to Various Decreases in Liver Function Tests and Is Dependent on the Underlying Polymorphism

In all patients, GGT was elevated at baseline and decreased significantly over 12 months, except in patients with the *ATP8B1* c. 2855 G>A (p.R952Q, rs765889649) polymorphism, which was present in only three patients. ALP, a sensitive marker of bile duct epithelial damage, was significantly reduced in patients with ATP8B1 c. 2855 G>A and *TM6SF2*, but not in patients with *ATP8B1* c. 2855 G>A.

In addition, liver damage as measured by the ALT level was significantly reduced in all polymorphisms except *TM6SF2*. AST showed no difference in any group but in the whole cohort (Figure 2).

In *ABCB4* c.504 C>T (p. N168=, rs1202283), GGT was reduced in patients with homozygous or heterozygous polymorphism, but in patients with polymorphism in ABCB4 c.A711 A>T, GGT was significantly reduced only in homozygous patients. In all patients with ABCB4 polymorphism, ALT was significantly reduced only in patients with homozygous polymorphisms (Figure 3).

All patients with polymorphisms in *ABCB11*, homozygote or heterozygote, showed a significant decrease in GGT. ALT was significantly reduced in patients with homozygous polymorphism in *ABCB11* c. 3084 A>G (p. A1028=, rs497692), and homozygous polymorphism in *ABCB11* 1331T>C (p.A444V, rs2287622) (Figure 4). In patients with homozygous polymorphism in *ABCB11* c. 3084 A>G, there was a decrease in AST after 12 months (p = 0.043, 42.6 (24–104) U/l vs. 30 (19–66) U/l), which was not significant in patients with polymorphism in *ABCB11* c. 1331 T>C (p = 0.13, 43.2 (23–69) U/l vs. 32 (24–79.8) U/l).

Patients with homozygote polymorphism in the *PNPLA3* showed a significant decrease in ALT and GGT under the administration of UDCA (Figure 5).

When regarding all patients with a polymorphism in any bile acid transporter gene, *TM6SF2* and *PNLPA3*, GGT, ALT, ALP, and AST were reduced.

### 3.5. UDCA Therapy Has No Effect on the Progression of Fibrosis

No significant difference was observed in liver fibrosis or steatosis, as measured by transient elastography (TE) or with Fib-4 or NAFLD fibrosis score (NFS), when grouped by polymorphism.

### 3.6. The Combination of Different Polymorphisms Shows the Highest Effect on ALT: GGT and Fib-4

To determine whether there was a potential cumulative effect of multiple polymorphisms, we calculated the absolute and relative decrease in ALT, GGT, and Fib-4 for each group per number of polymorphisms between the baseline and 12 months. The greatest decrease in ALT and Fib-4 was observed in patients with three concurrent polymorphisms (mean decrease ALT 27.7 U/l (7–44.4); relative decrease ALT 38.3% (12–68.8%)). However, GGT levels decreased more, but not significantly, with each additional polymorphism (Appendix A). Because of the small number of patients per group and the unequal distribution, we combined patients with 1–2 polymorphisms and patients with 3–5 polymorphisms. When comparing the relative and absolute reduction at 12 months, no significant results were found between the two groups (Appendix A).

## 4. Discussion

In this retrospective study, we evaluated the effect of UDCA in patients with NASH and bile acid transporter gene polymorphisms after 6 and 12 months. In almost all patients, liver function tests improved after administration. However, there was no significant effect on fibrosis and steatosis measured by non-invasive testing.

In both homozygous and heterozygous patients, there was a significant reduction in GGT in most patients, as expected, regardless of the underlying specific polymorphism. Furthermore, UDCA therapy resulted in reduced ALT levels in homozygous patients with *ABCB4*, *ABCB11*, and *PNPLA3* polymorphisms. Therefore, we suggest that patients with a homozygous polymorphism in any bile acid transporter gene may benefit from UDCA therapy in terms of liver damage.

All patients studied (who initially presented with elevated GGT levels) had polymorphisms in *ABCB4* or *ABCB11*. In the population-based Icelandic Genome Study, 0.2% of participants had a polymorphism in this gene which was associated with cirrhosis and elevated serum levels of liver-related biomarkers, ALT, AST, and GGT [25]. Notably, in our cohort, we only evaluated data from preselected patients (NAFLD and a high level of cholestatic parameters), hence the data are not comparable to a population study.

Previously, it has been described that ABCB11 c.1331T>C is associated with biopsy-proven liver fibrosis and cirrhosis in patients with HCV but not in NAFLD [26]. In addition, a retrospective study observed a reduction in HCC in patients with HCV cirrhosis treated with UDCA for five years, probably due to the anti-inflammatory effect of UDCA [27]. In this cohort, a significant reduction in ALT levels was observed in patients with *ABCB11* c.1331 T>C. Another study evaluating liver fibrosis by TE did not show a significant association in patients with chronic hepatitis C [28]. In agreement with the results of Iwata et al. [26], patients with the *ABCB11* c.1331 T>C polymorphism did not show a significantly higher level of fibrosis, as measured by surrogate parameters, compared to the wild type in our cohort. However, they were not matched to healthy controls. In an underlying *ABCB11* c.1331 T>C polymorphism, UDCA therapy does not seem to affect fibrosis, but ultimately there is a reduction in inflammation.

In a large Icelandic population study, *ABCB4* polymorphisms were not only associated with rare monogenic liver diseases but also with chronic liver diseases [12,25]. *ABCB4* c.523T>C was found to be associated with increased liver stiffness, whereas in *ABCB4* c.711A>T, it was found to be associated with increased hepatic stiffening only in the presence of polymorphism in *PNLAP3* [17]. In another study including 227 patients, this was an independent risk factor for fibrosis (as measured by TE and Fib-4) [16]. Most of the analyzed patients in our cohort showed no or mild fibrosis (F0-F1). Hence, we cannot make any conclusion if BA transporter gene polymorphisms are an independent risk factor for fibrosis in NASH patients. In addition, there was no difference in fibrosis in our cohort after one year of UDCA therapy. However, the ALT level decreased and steatosis, as measured by CAP and Fib4, decreased but not significantly (p = 0.175 (CAP), 0.054 (Fib-4)) in the homozygote group. This may be due to the small cohort of patients and the short observation period. Probably there is a long-term effect of UDCA administration, as suggested in a case report by Frider et al. [29].

*ABCB4* c. 504 A>T has only recently been discovered. It is mostly described in ICP. In one study, a patient with NAFLD who did not respond to therapy was genotyped and showed compound heterozygosity, including this polymorphism [30]. In this study, UDCA showed no difference in fibrosis or steatosis but a reduction in ALT and GGT level. As UDCA is well evaluated to decelerate the progress of MDR-3-associated diseases [13], it is possible that it could delay fibrosis in patients with NASH and *ABCB4* polymorphism.

Following the human genome project, polymorphisms associated with NAFLD have been described. Single-nucleotide polymorphism in *PNPLA3* (C>G, p. I148M) modifies NAFLD progression, is associated with the severity of steatohepatitis, severe fibrosis, and confers with an increased risk of developing HCC [31,32,33]. In Europeans, a heterozygote polymorphism has been described in 43.6–45% and a homozygote polymorphism in 12.1–25.5% of patients with NASH or NAFLD. Higher frequencies are described in Hispanic and lower frequencies in African American cohorts [31,34]. Lately, it has been shown that the incidence of liver-related events is higher in non-obese women in patients with *PNPLA3* polymorphism [35]. In this study cohort of 28 people, 21.4% of patients were heterozygous for the polymorphism in *PNPLA3* and 28.57% were homozygous. This can be explained by the fact that this is a small subset of patients who have already progressed to NASH and have high cholestatic parameters. This aspect must be considered in all the data presented here. In addition, *PNLPA3* sequencing was only available in 48.72% of patients. ALT and GGT decreased after treatment with UDCA. Given the high association of polymorphisms in *PNPLA3*, especially in non-obese women, with liver-related events, treatment with UDCA may be an affordable adjunct to more specific gene therapy, which is being evaluated in preclinical studies [35,36]. As each patient with a *PNPLA3* polymorphism had at least one additional polymorphism in one of the bile acid transporter genes, the implications of the findings need to be evaluated separately.

In a meta-analysis, UDCA treatment showed a significant effect on ALT and GGT in patients with liver disease, but in studies including only NAFLD or NASH patients, no significance was shown. Therefore, we believe that UDCA treatment may be beneficial only for patients with a bile acid transporter gene polymorphism [37]. In this study, we were able to show that 12 months of UDCA treatment resulted in a significant reduction in liver injury, as measured by ALT levels in patients with homozygote polymorphism in a BA transporter gene and NASH.

*TM6SF2* c.449 C>T is associated with increased hepatic triglyceride accumulation and hepatic steatosis, fibrosis, cirrhosis, and HCC [38,39]. The frequency was higher in European individuals (7.2%) than in other ethnic groups. Comparably, in this study, heterozygote polymorphisms were found in 7.7% of patients, but less than half of the cohort was tested (43.59%). Data on decreases in GGT or ALT levels in this cohort are not representative as there are only two patients available. In the literature, *TM6SF2* polymorphism is associated with an increase in the ALT level [38].

Because of the high concordance of NASH-related and bile acid gene polymorphisms leading to increased fibrosis, as described in several case reports [29], it is possible that some polymorphisms only reach significance when they are a haplotype. However, due to the small number of patients, further studies are needed to investigate whether patients with certain haplotypes and NASH benefit from UDCA therapy.

T2DM is an independent risk factor for NASH in NAFLD and the prevalence of T2DM in NAFLD and NASH patients is estimated to be estimated to be 22.51% and 43.63% [1]. In this cohort, T2DM was present in only 17.9% of patients, which would be expected to be higher. It may be that the progression to NASH is also influenced by a pathophysiological mechanism other than T2DM, hence the BA transporter gene polymorphism.

As described above, the conclusions of this study are mainly limited by the small cohort of patients. In addition, the patients were not matched with NASH or healthy controls. In addition, the data could not be stratified for confounders such as diabetes or weight loss because these data were not available. Weight loss and antidiabetic drugs are part of the NAFLD therapy and can eventually lead to a resolution of NASH [40] and are therefore a confounding factor regarding ALT levels. Endpoints for drug approval by EMA and FDA are the histological resolution of NASH and/or improvement in fibrosis as fibrosis is the most predictive factor [41]. Histological data were not available in this cohort; moreover, no effect could be shown on fibrosis using non-invasive tests.

Due to the limitations of the study (retrospective, small number of patients), the results should be applied with caution in clinical practice. However, we believe that in patients with high GGT and NAFLD, sequencing for polymorphism may be performed in individual cases where standard of care does not provide an appropriate result. Prospective data should be collected for further application in clinical practice.

In conclusion, therapy with UDCA in NASH patients may lead to a reduction in GGT and ALT levels when there are polymorphisms in the genes encoding bile acid transporters and regulation. This study did not describe a difference in clinical outcomes such as progression to cirrhosis or death. Especially in patients with lean NAFLD and no response to therapy, a bile acid transporter gene polymorphism could be considered as the underlying pathophysiology.

We believe that our retrospective findings need to be evaluated in prospective studies enrolling a larger number of patients. Furthermore, the underlying mechanism of how UDCA may reduce liver injury in NAFLD patients with derivations in the bile acid transport system needs to be investigated.

## Figures and Tables

**Figure 1 jpm-13-01180-f001:**
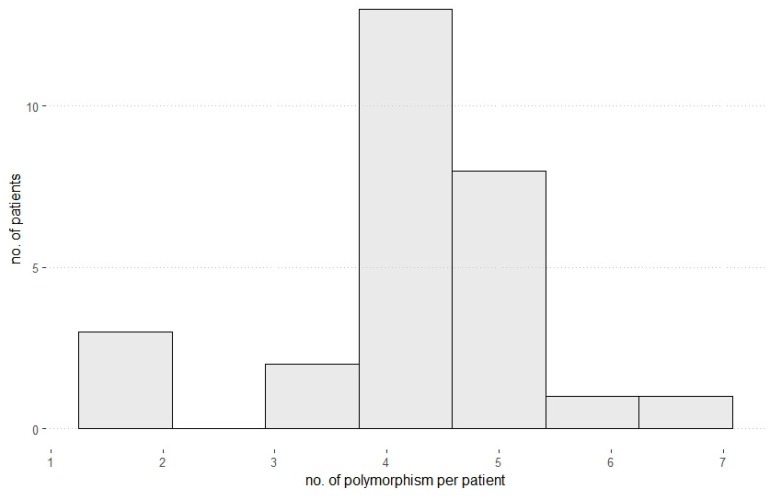
Number of polymorphisms per patient. Most patients showed 4 or 5 polymorphisms regarding all analyzed bile acid transporter genes and NAFLD-associated genes.

**Figure 2 jpm-13-01180-f002:**
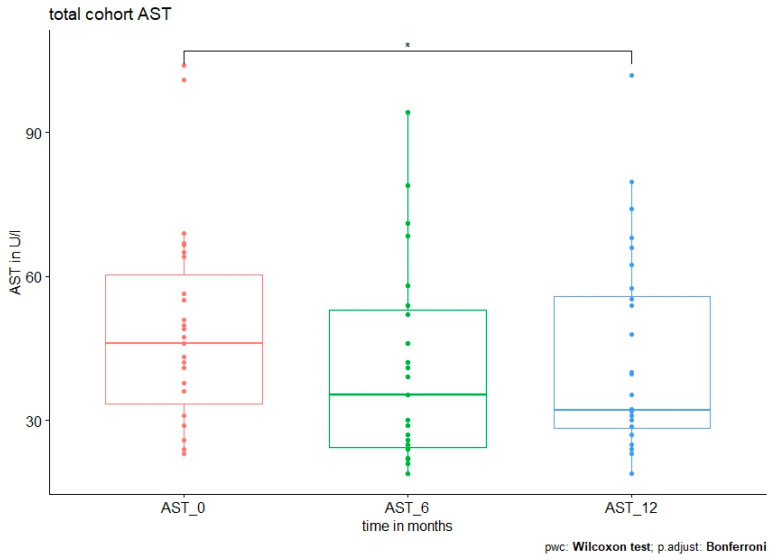
AST level at baseline and after 6 and 12 months in all patients. Median AST level was 46 (23–104) U/l at baseline vs. 35.4 (19–94.2) U/l at 6 months, *p* = 0.031. No further significant reduction was observed after 12 months (32.3 (19–102) U/l, *p* = 0.546 (0 vs. 12 months) and *p* = 1 (6 vs. 12 months).

**Figure 3 jpm-13-01180-f003:**
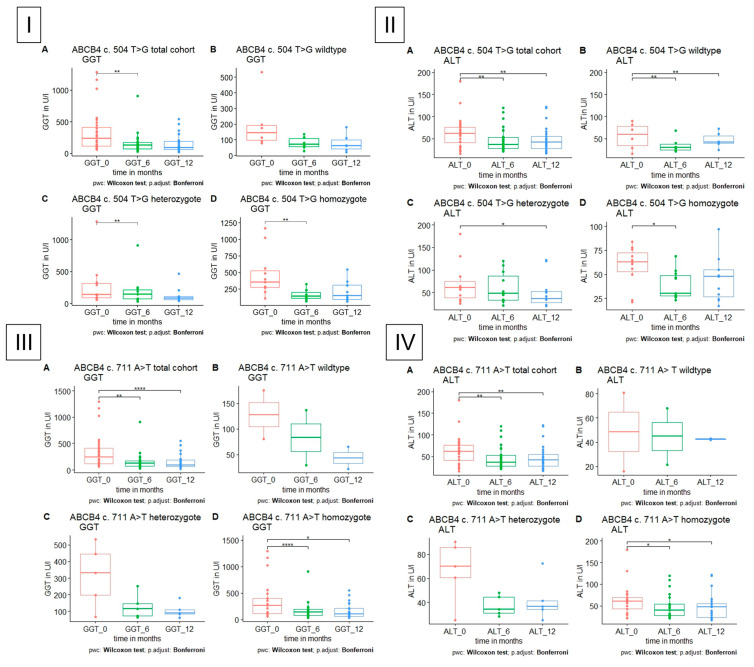
GGT and ALT level in patients with polymorphism in *ABCB4*, respectively, *A* total cohort, *B* wildtype, *C* heterozygote, *D* homozygote. (**I**) GGT is significantly reduced in heterozygote and homozygote polymorphism in *ABCB4* c.504 C>T after 6 months. (**II**) ALT is significantly reduced in heterozygote and homozygote polymorphism in *ABCB4* c.504 C>T after 6 and 12 months, respectively. (**III**,**IV**) GGT and ALT are significantly reduced in homozygote polymorphism in *ABCB4* c. 711 A>T after 6 and 12 months.

**Figure 4 jpm-13-01180-f004:**
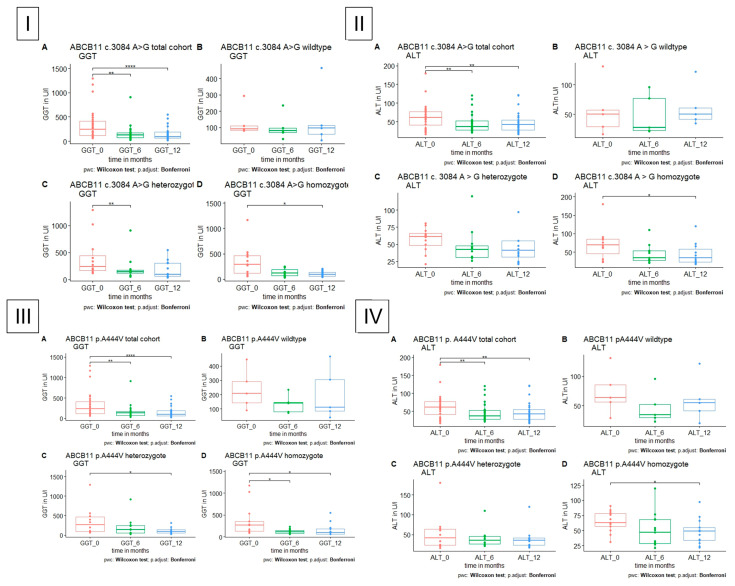
GGT and ALT level in patients with polymorphism in *ABCB11*, respectively, *A* total cohort, *B* wildtype, *C* heterozygote, *D* homozygote. (**I**) GGT is significantly reduced in heterozygote and homozygote polymorphism in *ABCB11* c. 3084 A>G after 6 months in heterozygote and after 12 months in homozygote patients. (**II**) ALT is significantly reduced in homozygote polymorphism in *ABCB11* c.3048 A>G after 12 months. (**III**) GGT is significantly reduced in heterozygote and homozygote polymorphism in *ABCB11* c. 1331 T>C after 12 months in heterozygote and after 6 and 12 months in homozygote patients. (**IV**) ALT is significantly reduced in homozygote polymorphism in *ABCB11* c. 1331 T>C after 12 months.

**Figure 5 jpm-13-01180-f005:**
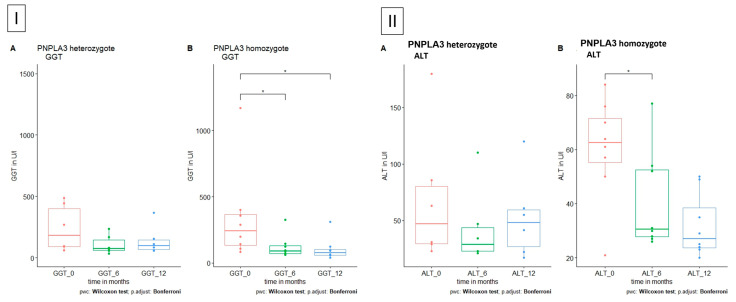
GGT and ALT level in patients with polymorphism in *PNPLA3*, respectively, *A* heterozygote and *B* homozygote. (**I**) GGT is significantly reduced in homozygote polymorphism in *PNPLA3* after 6 and 12 months. (**II**) ALT is significantly reduced in homozygote polymorphism in *PNPLA3* after 6 months.

**Table 1 jpm-13-01180-t001:** Description of polymorphisms.

Gen	SNP Cluster ID	c.	p.
*ABCB4*	rs45575636	c.1769G > A	p. R590Q
	rs1202283	c.504 C > T	p. N168=
	rs58238559	c.523 T> C	p. T175A
	rs2109505	c.711A>T	p. I237=
*ABCB11*	rs72549402	c.1445A>G	p. D482G
	rs497692	c. 3084 A>G	p. A1028=
	rs2287622	c.1331 T>C	p. A444V
*ATP8B1*	rs146599962	c.134A>C	p. N45T
	rs34018205	c.1286A>C	p. E429A
	rs121909100	c.1982T>C	p. I661T
	rs765889649	c.2855G>A	p. R952Q
*PNPLA3*	rs738409	C>G	p. I148M
*TM6SF2*	rs58542926	c.449 C>T	p. Glu167Lys;
FXR	rs56163822	c. −1 G>T	Intron

**Table 2 jpm-13-01180-t002:** Clinical and demographic data. Data are displayed for all patients. In T2DM, vitamin D deficiency, anemia, iron deficiency, folate deficiency, osteopenia, and smoking data were only available for 27 patients, and high cholesterol only for 26 patients. Vitamin B 12 deficiency: serum vitamin B12 < 200 ng/mL, vitamin D deficiency: serum 25-OH vitamin D < 10 ng/mL, anemia: hemoglobin < 11.6 g/dL, high cholesterol: serum-LDL > 3 mmol/L, iron deficiency serum ferritin: <15 ng/mL, folate deficiency: serum folate: <5 ng/mL.

	Total Cohort	Min–Max (%)
Number	28	
Age (mean)	42.29	17–75
Sex (female)	13	
Height (cm)	171.68	132–192
Weight (kg)	80.39	35–115
BMI (kg/m^2^)	26.98	19–39
FAST Score	0.68	0.3–0.97
Fib4 Score	1.56	0.0–5.82
NFS	−2.53	−8.08–1.53
Adipositas		
Normal weight	8	32
Obese	12	42.86
Adipostias Grad I	5	17.86
Adipositas Grad II	2	7.14
Alcohol		
None	19	67.86
Rarely	8	32
Sometimes	1	3.57
Comorbidities (no. of patients, %)		
T2DM (n = 27, 96)	6	21.42
Arterielle hypertonie (n = 28, 100)	8	32
Cardiovascular disease (n = 28, 100)	2	7.14
Chronic kidney disease(n = 28, 100)	2	7.14
Vitamin B12 deficiency (n = 27, 96)	2	7.14
Vitamin D deficiency (n = 27, 96)	5	17.86
Anemia (n = 27, 96)	2	7.14
Iron deficiency (n = 27, 96)	2	7.14
Folat deficiency (n = 27, 96)	0	0
Osteopenia (n = 27, 96)	3	10.71
High cholesterol (n = 26, 92.86)	14	0.5
Smoking (n = 27, 96)	5	17.86

**Table 3 jpm-13-01180-t003:** Frequency of polymorphisms.

Polymorphism	Heterozygote	Homozygote	Wildtype	No Data
*ABCB4* c.504 C>T	11	11	6	0
*ABCB4* c.771 A>T	5	21	2	0
*ABCB11c*. 3084 A>G	12	11	5	0
*ABCB11* c. 1331 T>C	10	13	5	0
*PNPLA3* rs738409 C>G	6	8	0	14
*TM6SF2* c.449 C>T	2	0	10	16
FXR c. −1 G>T	3	0	25	0

## Data Availability

Data will be made available by the corresponding author upon reason-able request.

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
