# Peer review of "Influence of the Bile Acid Transporter Genes ABCB4, ABCB8, and ABCB11 and the Farnesoid X Receptor on the Response to Ursodeoxycholic Acid in Patients with Nonalcoholic Steatohepatitis"

_jpm, 2023, doi:10.3390/jpm13071180_

Round 1
Reviewer 1 Report
- New references for 2022, 2023 will be added to text
- Future aspects are missing
- The first time you include acronyms within the text, you have to write them in full. After that, you should report them as abbreviations only.
In the introduction, in the last paragraph, the importance of the issue and the necessity of implementation should be explained. Also explain why you did the present study
-The p values are given in results
in the work method; Was it random in collecting the samples? Sample size formula
The discussion needs to be fundamentally rewritten:
The results of other studies, whether similar or different, should be brought and compared and inferred, and what is the difference between the current article and those articles?
Mention each of your important findings, then compare with others and mention your interpretation and inference of the results and causes of similarities and differences.
Indicate the application of the results
The limitations of the study should be mentioned
Running title should be added
The type of study should be mentioned.
It should also be mentioned the sampling method and where and what center the patients were selected from.
On what basis and how was the sample size selected? The number of samples should be justified in a scientific way. Please, according to the main purpose of the study, add the appropriate formula and valid parameters by mentioning the source.
Wishing you all of the best
Yours sincerely
- New references for 2022, 2023 will be added to text
- Future aspects are missing
- The first time you include acronyms within the text, you have to write them in full. After that, you should report them as abbreviations only.
In the introduction, in the last paragraph, the importance of the issue and the necessity of implementation should be explained. Also explain why you did the present study
-The p values are given in results
in the work method; Was it random in collecting the samples? Sample size formula
The discussion needs to be fundamentally rewritten:
The results of other studies, whether similar or different, should be brought and compared and inferred, and what is the difference between the current article and those articles?
Mention each of your important findings, then compare with others and mention your interpretation and inference of the results and causes of similarities and differences.
Indicate the application of the results
The limitations of the study should be mentioned
Running title should be added
The type of study should be mentioned.
It should also be mentioned the sampling method and where and what center the patients were selected from.
On what basis and how was the sample size selected? The number of samples should be justified in a scientific way. Please, according to the main purpose of the study, add the appropriate formula and valid parameters by mentioning the source.
Wishing you all of the best
Yours sincerely
Reviewer 2 Report
I have studied carefully the manuscript entitled "Influence of the Bile Acid Transporter Genes ABCB4, ABCB8 and ABCB11 and the Bile Acid Receptor FXR on the Response to UDCA in Patients with Nonalcoholic Steatohepatitis" by Kreimeyer H. et al.
The manuscript carries some novelty as there is no publication referring to the effect of ABCB4, ABCB11, TM6SF2, NR1H4, ATP8B1and PNPLA3 in the treatment of non-alcoholic steatohepatitis (NASH) with ursodeoxycholic acid (UDCA).
Before considering publication, the authors are encouraged to discuss the following issues:
Major issue
1. The non-parametric Friedman One-Way Repeated Measures Analysis of Variance by Ranks, used to compare three or more matched groups, is preferrable to multiple Wilcoxon tests with Bonferroni correction in all comparisons depicted at Figures 2, 3, 4, and 5. Interestingly, the Friedman test is referred at the "Methods" section.
2. There is a substantial amount of data mising concerning PNPLA3 rs738409 C>G and TM6SF2 c.449 C>T polymorphisms (in 14 and 16 patients, respectively), the authors are suggested to revise Figure 1 so as to depict olny polymorphisms with no missing data (ABCB4 c.504 C>T, ABCB4 c.771
A>T, ABCB11c. 3084 A>G, ABCB11 c. 1331 T>C, and FXR c. -1 G>T. Furthermore, the authors could introduce a properly designed statistical test in order to describe the potential cumulative effect of all polymorphisms in GGT, ALT, and Fib-4 score.
3. The references are outdated, since dozens of relevant publications have been available within 2023 (see e.g. i) Lindén D. et al. Therapeutic opportunities for the treatment of NASH with genetically validated targets. J Hepatol. 2023 May 17:S0168-8278(23)00335-5.; ii) Rosso C. et al. Impact of PNPLA3 rs738409 Polymorphism on the Development of Liver-Related Events in Patients With Nonalcoholic Fatty Liver Disease. Clin Gastroenterol Hepatol. 2023 May 4:S1542-3565(23)00324-5.; iii) Tacke F. et al, An integrated view of anti-inflammatory and antifibrotic targets for the treatment of NASH. J Hepatol. 2023 Apr 14:S0168-8278(23)00218-0).
Minor issues
1. As non-parametric analysis has been followed, medians along with either interquartile range or range would be preferrable to means.
2. Figures 3 and 4 contain 16 different charts each. The authors are wellcome to improve their quality in order to be easily readable.
Minor editing of English language is required.
Round 2
Reviewer 2 Report
I have studied the revised version of the manuscript entitled "Influence of the Bile Acid Transporter Genes ABCB4, ABCB8 and ABCB11 and the Bile Acid Receptor FXR on the response to UDCA in Patients with Nonalcoholic Steatohepatitis" by Kreimeyer Henriette et al.
The authors have made considerable effort to effectively respond to the queries raised by the reviewers. The text of the revised manuscript has been ameliorated, the methodology has been improved, and the references added have substantially contributed to rendering the literature used up-to-date.
Under the above-mentioned circumstances, I have no further major objection concerning publication.